# Feature Distillation is the Better Choice for Model-Heterogeneous Federated Learning

**Yichen Li**[1,2], **Xiuying Wang**[3], **Wenchao Xu**[4] **Haozhao Wang**[1],
**Yining Qi**[1], **Jiahua Dong**[2], **Ruixuan Li**[1*]
[1]School of Computer Science and Technology,
Huazhong University of Science and Technology, Wuhan, China
[2]Mohamed bin Zayed University of Artificial Intelligence, Abu Dhabi, United Arab Emirates
[3]International School, Beijing University of Posts and Telecommunications, Beijing, China
[4]Department of Computing, The Hong Kong Polytechnic University, Hong Kong, China
{ycli0204,hz_wang}@hust.edu.cn, leowang980@bupt.edu.cn

## Abstract

Model-Heterogeneous Federated Learning (Hetero-FL) has attracted growing attention for its ability to aggregate knowledge from heterogeneous models while keeping private data locally. To better aggregate knowledge from clients, ensemble distillation, as a widely used and effective technique, is often employed after global aggregation to enhance the performance of the global model. However, simply combining Hetero-FL and ensemble distillation does not always yield promising results and can make the training process unstable. The reason is that existing methods primarily focus on logit distillation, which, while being model-agnostic with softmax predictions, fails to compensate for the knowledge bias arising from heterogeneous models. To tackle this challenge, we propose a stable and efficient _Feature_ _D_istillation for model-heterogeneous _Fed_erated learning, dubbed **FedFD**, that can incorporate aligned feature information via orthogonal projection to integrate knowledge from heterogeneous models better. Specifically, a new feature-based ensemble federated knowledge distillation paradigm is proposed. The global model on the server needs to maintain a projection layer for each client-side model architecture to align the features separately. Orthogonal techniques are employed to re-parameterize the projection layer to mitigate knowledge bias from heterogeneous models and thus maximize the distilled knowledge. Extensive experiments show that FedFD achieves superior performance compared to state-of-the-art methods.

## 1 Introduction

Federated Learning (FL) has become a core method for collaboratively training neural networks across distributed clients while ensuring data privacy is protected [34, 50, 29]. Recently, this framework has attracted considerable attention and is being applied in various domains, such as autonomous driving systems [32, 37], recommender systems [27, 26], and intelligent healthcare solutions [7, 40].

Typically, FL has been actively studied with a homogeneous model setting. With the development of IoT products, different devices often possess varying computing resources, namely different model training capabilities [39, 31]. In such a scenario, it is essential to explore model-heterogeneous federated learning (Hetero-FL) to maximize the utilization of distributed computing resources. The key challenge here lies in how to aggregate the shared knowledge from different models.

---

*Ruixuan Li is the corresponding author.

39th Conference on Neural Information Processing Systems (NeurIPS 2025).

To address this issue, knowledge distillation [15] has emerged as a promising approach, focusing on aggregating the output soft predictions of multiple local models into the global model. The authors in [52] propose aggregating the logit output instead of model parameters and [21] utilize the aggregated class score to regulate the local training process. Building on the partial parameter aggregation proposed in [6], many works use the ensemble distillation to improve the global model like [33, 49]. [48] and [55] have focused on knowledge selection with logit distillation to optimize the aggregated model. [43] identifies the accurate and precise knowledge from local and ensemble predictions.

Although these methods achieve performance gains by integrating the ensemble distillation technique with Hetero-FL, how this distillation technique works in Hetero-FL remains uncertain, especially when it comes to logit-based distillation. While logit distillation can theoretically address the shifts in the class probability distribution caused by data heterogeneity in FL with homogeneous models perfectly by using a publicly available dataset with a similar distribution [46], *it is a sub-optimal strategy for model heterogeneity where the key challenge lies in different hidden layer representations.* During the distillation, each

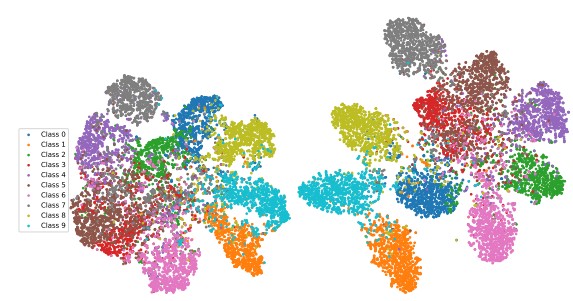

(a) Logit Representation.    (b) Feature Representation.

Figure 1: The t-SNE visualization of ensemble knowledge representation by aggregated heterogeneous models on CIFAR-10.

heterogeneous model will map the sample into a distinct feature space, resulting in significant variations in their softmax predictions (logits) [10]. It is widely acknowledged that the logit representation only focuses on the output layer but *can not align the representation in different feature spaces of heterogeneous models well, decreasing the distillation effectiveness.*

We use t-SNE [45] to visualize the ensemble knowledge representation from client-side models on the distillation dataset in Figure.1. Compared with feature representation in our method, aggregated logit representation has **fuzzy** classification boundaries, indicating that the performance of the teacher model is not promising enough. *Moreover, we empirically find that the logit distillation will cause an unstable federated training process, then directly aggregating logits as the teacher's prediction is not always effective.* The specific experiments will be provided in Section 3.2.

To break the limitations of logit distillation for FD with heterogeneous models, we in this paper explore feature distillation to ensemble the distilled knowledge where the feature representation is closely related to the model structure. Using feature representation for ensemble distillation in Hetero-FL poses a novel challenge, as in a centralized environment, feature knowledge is often distilled from a large model (teacher model) to a smaller model (student model) with an external projection layer [41, 35]. This differs from the Hetero-FL, where small models on various client sides usually ensemble distill knowledge to a large model on the server, and the structures of these client-side models vary. Thus, designing the utilization of the projection layer and aligning feature representations of different client-side models becomes a primary challenge in ensemble distillation for Hetero-FL.

To explore this idea, we propose a stable and effective feature distillation method for Hetero-FL named **FedFD** that can align feature representations with orthogonal projection to mitigate the knowledge bias aggregated from heterogeneous models. More specifically, for each distillation sample, clients will extract the feature representation, and the server aggregates the representations of the same client-side model architecture to obtain a feature cluster formed by aggregated features from different model architectures. For each representation within the feature cluster, the server needs to train a projection layer to align the extracted feature representation of the server model with it and optimize the parameters of both the projection layer and the feature extractor of the server model. Furthermore, to prevent knowledge bias among aggregated feature representations, we utilize orthogonal projection techniques to maximize the transferred knowledge.

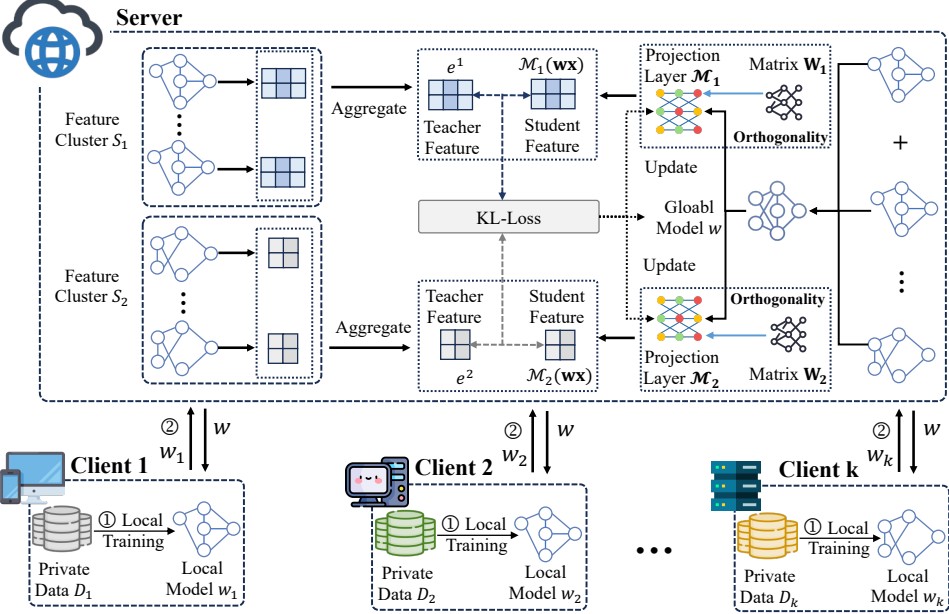

Figure 2: The framework of FedFD. Before knowledge distillation, each client first trains on its local dataset and then uploads its local model to the server. The server aggregates these models to obtain a global model. During the distillation process, clients perform hierarchical feature alignment. This involves firstly aggregating feature representations from client-side models with consistent architectures. Then, for each aggregated feature representation, a projection layer is maintained. This projection layer is obtained by transforming a square matrix to ensure its orthogonality. Finally, feature distillation is achieved by aligning the feature representations of different model architectures using KL divergence separately.

Through extensive experiments over three datasets and different settings (various model architectures and data distributions), we show that the proposed framework significantly improves the model accuracy as compared to state-of-the-art algorithms. The contributions of this paper are:

- We provide an in-depth analysis of model-agnostic federated knowledge distillation, identifying that existing methods primarily rely on logit distillation, which poses significant challenges for misleading distilled knowledge representation with heterogeneous models.

- We propose a novel framework named FedFD which can be seen as an off-the-shelf personalization add-on for Hetero-FL and it inherits privacy protection and efficiency properties as traditional distillation methods.

- We conduct extensive experiments on various datasets and settings. Experimental results illustrate that our proposed model outperforms the state-of-the-art methods by up to **16.09%** in terms of test accuracy on different tasks.

## 2 Related Work

**Federated Learning** is a framework that trains a unified global model by aggregating individual models from multiple clients, each trained on their locally stored datasets [25, 47, 28, 30]. One of the notable FL architectures is FedAvg [34], which strengthens the global model by aggregating parameters from locally trained models on private data. [23] involves incorporating a proximal term to mitigate the impact of differing data distributions across devices. While these methods are devoted to the homogeneous model across clients, [6] proposes to enable the training of heterogeneous local models with varying computation complexities on different clients. Similar to [6], [51] integrates flexibility in both width and depth, utilizing skip connections to bypass certain layers and structured pruning to manage width. [17] is a strategy that adjusts to varying depths by aggregating common layers from clients' networks, such as the VGG network, to create global models from different layer

groups. This paper focuses on federated learning with heterogeneous models across clients with improved knowledge distillation techniques.

**Knowledge Distillation** leverages the knowledge of a pre-trained model to supervise a smaller model, facilitating its application and deployment in environments with limited resources [14]. The domain primarily encompasses two areas: logits distillation [38, 18, 1, 16] and feature distillation [44, 36, 13, 5]. Logits distillation, centered on classification tasks, introduces an additional goal to minimize the prediction discrepancy between the student and teacher models, initially using KL divergence [14] and later extended through spherical normalization [8], label decoupling [57], and probability reweighing [38]. Our research focuses on feature distillation due to its versatility across tasks [5] and modalities [42]. The FSP matrix is manually designed to capture feature relationships across residual layers [54]. Similarly, other studies proposed transferring knowledge via Gram matrices [24]. Activation boundaries and gradients capturing the loss landscape have also proven effective as supervisory signals [44]. We in this paper particularly focus on the feature distillation in FD by orthogonal projection and ensemble distillation.

**Federated Distillation** involves extracting knowledge from multiple teacher models, each trained by different clients, and transferring it to a student model [53, 9, 2]. [33] introduces the concept of applying knowledge distillation in a server setting, utilizing an unlabeled proxy dataset to transfer knowledge from local models to a global model. [4] further develops this by linearly aggregating multiple local models, using weights derived from the Bayesian posterior, to create a series of combined models. These combined models are then distilled into a single global model. To break out the limitation of relying on an unlabeled auxiliary dataset, [58, 56, 49] propose suggest replacing the proxy dataset with data generated by generative models, enabling the distillation without the need for actual data. We analyze the challenge of employing existing FD methods in the FL with heterogeneous models and propose to develop feature distillation instead of logit distillation.

## 3 Methodology

In this section, we first formulate the standard FL process with both homogeneous and heterogeneous models. Then, we analyze the failure of logit distillation in Hetero-FL with experiments. Last, we introduce our feature distillation-based method, FedFD. The workflow of FedFD is shown in Algorithm 1 and Figure.2 illustrates the FedFD framework.

### 3.1 Problem Formulation

A typical FL problem can be formalized by collaboratively training a global model for $K$ total clients in FL. We consider each client $k$ can only access to his local private dataset $D_k = \{x_k^{(i)}, y_k^{(i)}\}$, where $x_k^{(i)}$ is the $i$-th input data sample and $y_k^{(i)} \in \{1, 2, \cdots, C\}$ is the corresponding label of $x_k^{(i)}$ with $C$ classes. We denote the number of data samples in dataset $D_k$ by $|D_k|$. The objective of the FL system is to learn a global model $w$ that minimizes the total empirical loss over the dataset $D$:

$$\min_w \mathcal{L}(w) = \sum_{k=1}^{K} \frac{|D_k|}{|D|} \mathcal{L}_k(w),$$

$$\text{where } \mathcal{L}_k(w) = \frac{1}{|D_k|} \sum_{i=1}^{|D_k|} \mathcal{L}_{CE}(w; x_i^k, y_i^k), \tag{1}$$

where $\mathcal{L}_k(w)$ is the local loss in the $k$-th client and $\mathcal{L}_{CE}$ is the cross-entropy loss function that measures the difference between the prediction and the ground truth labels. ***For homogeneous models***, the server only needs to average local parameters to obtain the global follow:

$$w := \sum_{k=1}^{K} \frac{|D_k|}{|D|} \cdot w_k. \tag{2}$$

While ***for heterogeneous models***, we follow the protocol proposed in [6] and assume that all client models can be divided into $p$ different architecture sets $\{S_1, S_2, \ldots, S_p\}$. Then, we perform the

global aggregation as follows:

$$w_p^s = \sum_{w_i \in S_p} \frac{w_i}{|S_p|}, \;\; w_{p-1}^s \setminus w_p^s = \frac{1}{K - |S_p|} \sum_{w_i \in S_p} w_{p-1}^s \setminus w_p^s,$$

$$w = w_p^s \cup (w_{p-1}^s \setminus w_p^s), \ldots, \cup(w_1^s \setminus w_2^s). \tag{3}$$

where $w_p^s$ denotes the aggregated model of the $p$-th architecture of client-side models. For notational convenience, we have dropped the training iteration index and simplified the weight for each aggregated client-side model in Eq. (3), which is often weighed by the ratio of sample numbers.

## 3.2 Logit Distillation Fails in Hetero-FL

In existing FL methods, the key technique is to transfer knowledge from the teacher model to the student model by utilizing the model's soft predictions (logits) on the distillation dataset. Here, the teacher model prediction is defined with aggregated logits from multiple clients, which can also be referred to as ensemble distillation in FL. This technique works because in Homo-FL (for homogeneous models), where the client models share the same structure, the logits output by these models do not suffer from model-based biases and can mitigate further data heterogeneity. However, it is inadvisable to directly apply this logit distillation technique to Hetero-FL (for heterogeneous models), and existing related work has not addressed whether the logits from different client model architectures can be directly aggregated as the teacher model prediction. Based on Figure.1, we conduct further experiments. We selected several robust ensemble distillation methods and conducted experiments on CIFAR-10 under both Homo-FL and Hetero-FL settings. The details of these methods are described in Section 4.1, and the learning curves are shown in Figure.3.

Under the Homo-FL setting, all methods converge quickly and stably, demonstrating the effectiveness of logit distillation. However, in the Hetero-FL setting, the FL algorithm experiences training instability, where peaks repeatedly appear in the curve, and the methods ultimately fail to converge to the optimal value. Although logit distillation can accelerate model convergence in the early stages of training, as the local model gradually converges on local data, the distillation process becomes highly unstable. This is contrary to the results observed in Homo-FL. The reason is that logit distillation solely relies

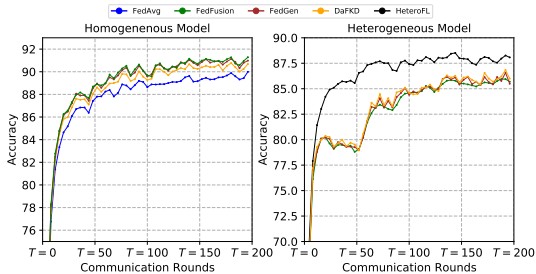

Figure 3: Learning curves of logit distillation-based methods on CIFAR-10 with different client-side model architectures.

on soft predictions without considering differences in model architectures. Therefore, we will next introduce our feature-based distillation method to address this issue.

## 3.3 FedFD: Feature Distillation for Hetero-FL

In this paper, we seek to unlock the potential of feature distillation in Hetero-FL. Despite the generality of feature distillation, it is frequently accompanied by complex design choices and heuristics. These decisions stem from loss functions between intermediate feature representations, leading to extra training costs. To overcome these limitations, we just employ the feature distillation that solely utilizes feature representation before the classifier.

In Hetero-FL, although the teacher model is typically the ensemble output of client-side small models, it is still necessary to attach a projection layer after the global model (student model) to ensure that knowledge can be back-propagated through the projection layer into the model parameters during the distillation process. However, this approach poses **two** challenges: (1) Due to the diversity of client-side model architectures, if the server maintains a personalized projection layer for each client, it may lead to difficult training because each client contributes only a tiny amount of knowledge. Additionally, in FL scenarios, the number of clients participating in pre-training is usually large, and maintaining an excessive number of projection layers can accumulate and result in significant storage costs for the server. (2) The feature knowledge derived from different client-side models may conflict

---
**Algorithm 1:** FedFD
___
**Input**   :$T$: communication round;  $K$: client number;  $D_k$: local dataset for the client $k$;  $w$: global model;  $w_k$: local model;  $\mathcal{M}_k$: projection layer;  $S_k$: the $k$-th model architecture; $\mathbf{W}_k$: matrix for orthogonality.

**Output**:$w, \{w_1, \ldots, w_k\}$: global and local models.

**1 for** $c = 1$ *to* $T$ **do**                                                              // communication round
**2**  |   Server randomly selects a subset of devices $S_t$;
**3**  |   Server send the global model $w$ to devices.
**4**  |   **for** *each selected client* $k \in S_t$ **in parallel do**
**5**  |   |   Train the local model $w_k$ with (1);
**6**  |   |   Send the local model $w_k$ back to the server.
**7**  |   **end**
**8**  |   $w \leftarrow$ ServerAggregation($\{w_k\}_{k \in S_t}$) with (3);
**9**  |   Get the aggregated feature representation $e^d$ with (5);
**10** |   Orthogonalize $\mathbf{W}_D$ to obtain projection layer $\mathcal{M}_d$ with (7);
**11** |   Distill the feature knowledge to the global model with (9).
**12 end**
___

within different projection layers, and combining the complex non-linear knowledge together may not optimally train the feature extractor module of the global model.

To address these issues, we propose the feature-based distillation method, FedFD, with two main components: hierarchical feature alignment and parameter orthogonality. To obtain the classification model, in each round, the participating client $k$ firstly locally trains the model $w_k$ with (1) and sends it to the server. After receiving uploaded models, the server aggregates multiple local models to get the global model $w$ with (3). Like [33][49][58], the server treats the knowledge of the client model on the distillation dataset as a teacher model to distill the global model and enhance its performance. Based on previous experiments, we explore using feature representation as the distilled knowledge. Denote that the local model $w_k$ of client $k$ consists of a feature extractor $\mathbf{w}_k^d$ and a classifier head $\theta_k$. $d$ is the dimension of the output feature. For each distillation sample $\mathbf{x}$, its feature representation $e_k^d$ over the extractor $\mathbf{w}_k^d$ can be obtained as follows:

$$e_k^d = f(\mathbf{w}_k^d; \mathbf{x}), \ \ \forall k \in [1, K]. \tag{4}$$

Then, we divide the features into $m$ groups $\{S_{d_1}, \ldots, S_{d_m}\}$, where each group $S_d = \{\mathbf{w}_1^d, \ldots, \mathbf{w}_k^d\}$ contains all extractors with the same structure outputting the $d$-dimensional feature. We use $|S_d|$ to represent the number of features in the group $S_d$. Next, we aggregate all feature representations in the group $S_d$ as:

$$e^d = \frac{1}{|S_d|} \sum_{i=1}^{|S_d|} e_i^d, \ \ \text{where} \ \ S_d = \{\mathbf{w}_1^d, \ldots, \mathbf{w}_{|S_d|}^d\}. \tag{5}$$

Instead of maintaining the projection layer separately for each client, the server now only needs to train $(m-1)$ projection layers $\{\mathcal{M}_2, \ldots, \mathcal{M}_m\}$. *This not only reduces training parameters but also ensures that each projection layer has sufficient knowledge for distillation.*

However, this does not resolve the knowledge conflict among different model architectures, as the server still needs to integrate knowledge from various projection layers to optimize the global model parameters $\mathbf{w}$. We indicate the knowledge $\mathbf{S}_d$ after the projection layer $\mathcal{M}_d$ as:

$$\mathbf{S}_d = \mathcal{M}_d(\mathbf{w}\mathbf{x}) = \mathcal{M}_d\mathbf{Z}, \ \ \text{where} \ \ \mathbf{Z} = \mathbf{w}\mathbf{x}. \tag{6}$$

While in knowledge distillation, a larger number of distillation samples often leads to better distillation effects, as they can better cover the knowledge within the training data. This results in $\mathbf{Z}$ being a full-rank matrix; consequently, the $\mathbf{S}_d$ will also be a full-rank matrix that can be regarded as a non-linear mapping process, where the global model fails to discern the knowledge, leading to knowledge conflicts.

To address this issue, since we cannot alter the knowledge distribution of $\mathbf{Z}$, we choose to process the projection layer $\mathcal{M}_d$ so that they can map the knowledge $\mathbf{Z}$ into separate feature spaces. Here, we

will investigate orthogonal projection transformations, which can resolve knowledge conflicts and maintain feature shapes due to their rotational invariance. It is noteworthy that, due to the differing feature dimensions between $\mathbf{Z}$ and $\mathbf{S}_d$, with $\mathbf{Z}$ having a higher dimension than $\mathbf{S}_d$, $\mathcal{M}_d$ is no longer an orthogonal square matrix but rather a column matrix, specifically a Stiefel matrix manifold with orthogonal column vectors [12]. The favorable topological properties of such a matrix can ensure support for gradient descent updates.

To maintain the orthogonality of the column vectors in $\mathcal{M}_d$, we have observed many existing methods such as Cayley transformation [3] and Gram-Schmidt method. However, these methods all come with relatively high time performance costs. In this paper, we propose to generate the $\mathcal{M}_d$ with a skew-symmetric matrix $\mathbf{W}_d$ as follows:

$$\exp(\mathbf{W}_d) \cdot \exp(\mathbf{W}_d)^T = \exp(\mathbf{W}_d + \mathbf{W}_d^T) = \exp(-\mathbf{W}_d^T + \mathbf{W}_d^T) = \mathbf{I}. \tag{7}$$

$$\exp(\mathbf{W}_d) = \mathbf{I} + \mathbf{W}_d + \frac{\mathbf{W}_d^2}{2!} + \frac{\mathbf{W}_d^3}{3!} + \cdots + \frac{\mathbf{W}_d^n}{n!}. \tag{8}$$

where $\mathbf{W}_d = -\mathbf{W}_d^T$. The parameters of $\mathrm{W}_d$ are randomly initialized. Although this is an infinite series, a reasonably accurate value can usually be obtained by taking the first few terms at a very low computational cost. Ultimately, $\mathcal{M}_d$ can be obtained by truncating the column vectors of $\exp(\mathbf{W}_d)$ to match the feature dimension of $\mathbf{S}_d$. The client updates $\mathbf{W}_d$ through back-propagation with Eq.(9). *By employing orthogonal projection, we ensure linear transformations of features, preventing knowledge conflicts and preserving feature shapes, thereby achieving maximized knowledge distillation.*

Through orthogonal projection, $\mathcal{M}_d(\mathbf{wx})$ and $e^d$ share the same dimensionality. Next, we update the parameters $w$ and $\mathcal{M}$ to align the feature representations using Kullback-Leibler divergence:

$$\min_{\mathbf{w}, \{\mathcal{M}_2, \ldots, \mathcal{M}_m\}} \frac{1}{m-1} \sum_{i=2}^{m} KL(\mathcal{M}_i(\mathbf{wx}), e^i). \tag{9}$$

After knowledge distillation, the server will broadcast the global model $w$ to the clients participating in the following communication round.

**Modularity.** FedFD demonstrates a key characteristic: modularity. Existing FL techniques can be seamlessly integrated with FedFD as a ready-to-use enhancement with the following several benefits:

- **Optimization:** The proposed framework can accommodate aggregation methods beyond HeteroFL [6] for global model updates, retaining convergence advantages.
- **Privacy:** FedFD maintains the same level of network communication as standard FL algorithms, avoiding privacy issues associated with uploading generative models or extra prototype-like information.
- **Flexibility:** Although we search for the additional distillation datasets here, our framework can be combined with existing data-free distillation techniques, balancing resource constraints and computational overheads in selecting distillation datasets.

## 4 Experiments

In this section, we evaluate our proposed method using three datasets and various baselines. We investigate the relationship between data heterogeneity and training efficiency. Additionally, we conduct ablation studies to examine each module in FedFD. Finally, we conduct a sensitivity analysis to verify the effectiveness of our method.

### 4.1 Experiment Setup

**Dataset:** We conduct our experiments with heterogeneously partitioned datasets over three datasets: CIFAR-10, CIFAR-100 [19], and Tiny-ImageNet [20]. Like [58, 49], we use the Dirichlet distribution $Dir(\alpha)$ on labels to simulate the data heterogeneity. We apply all the training samples and distribute them to user models, and we use all the testing samples for the performance evaluation.

**Baselines:** For a fair comparison with other key works, we follow the same protocols proposed by [6] to set up FD tasks with heterogeneous models, which is recorded as "-hetero". We evaluate our

Table 1: Performance comparison of various methods with the test accuracy.

| Categories | Methods | Metrics | CIFAR-10 | | | CIFAR-100 | | | Tiny-ImageNet | | |
|---|---|---|---|---|---|---|---|---|---|---|---|
| | | | $\alpha$=10.0 | $\alpha$=1.0 | $\alpha$=0.1 | $\alpha$=10.0 | $\alpha$=1.0 | $\alpha$=0.1 | $\alpha$=10.0 | $\alpha$=1.0 | $\alpha$=0.1 |
| Classic FL | HeteroFL | Local | $80.11_{\pm 0.95}$ | $75.83_{\pm 1.35}$ | $63.53_{\pm 4.66}$ | $53.37_{\pm 1.11}$ | $49.33_{\pm 1.76}$ | $38.07_{\pm 4.52}$ | $29.71_{\pm 2.85}$ | $24.12_{\pm 5.03}$ | $18.54_{\pm 3.40}$ |
| | | Global | $88.45_{\pm 0.03}$ | $87.53_{\pm 0.15}$ | $78.02_{\pm 0.65}$ | $58.51_{\pm 0.19}$ | $57.42_{\pm 0.12}$ | $53.98_{\pm 0.43}$ | $32.38_{\pm 0.51}$ | $29.88_{\pm 2.72}$ | $23.25_{\pm 2.96}$ |
| | MOON-hetero | Local | $80.53_{\pm 0.77}$ | $75.87_{\pm 0.99}$ | $63.98_{\pm 3.20}$ | $52.85_{\pm 2.03}$ | $50.21_{\pm 2.42}$ | $38.21_{\pm 3.61}$ | $28.36_{\pm 1.70}$ | $24.47_{\pm 2.42}$ | $17.94_{\pm 2.51}$ |
| | | Global | $88.05_{\pm 0.14}$ | $87.92_{\pm 0.17}$ | $79.05_{\pm 0.89}$ | $58.39_{\pm 0.20}$ | $57.55_{\pm 0.17}$ | $55.01_{\pm 0.34}$ | $31.28_{\pm 1.61}$ | $30.68_{\pm 1.92}$ | $24.91_{\pm 1.78}$ |
| Homo-FL | FedFusion-hetero | Local | $82.35_{\pm 0.51}$ | $75.27_{\pm 1.00}$ | $62.20_{\pm 5.37}$ | $51.23_{\pm 1.73}$ | $48.21_{\pm 2.08}$ | $37.53_{\pm 5.15}$ | $33.98_{\pm 2.07}$ | $25.62_{\pm 2.29}$ | $20.31_{\pm 5.25}$ |
| | | Global | $86.69_{\pm 0.30}$ | $85.70_{\pm 0.15}$ | $78.47_{\pm 1.11}$ | $59.53_{\pm 0.11}$ | $58.86_{\pm 0.12}$ | $56.12_{\pm 0.35}$ | $35.05_{\pm 1.87}$ | $31.56_{\pm 1.23}$ | $26.09_{\pm 0.73}$ |
| | FedGen-hetero | Local | $82.29_{\pm 0.62}$ | $76.01_{\pm 0.97}$ | $62.74_{\pm 3.17}$ | $51.99_{\pm 3.67}$ | $47.80_{\pm 1.12}$ | $39.79_{\pm 6.02}$ | $34.04_{\pm 2.93}$ | $25.37_{\pm 2.56}$ | $21.77_{\pm 6.89}$ |
| | | Global | $87.34_{\pm 0.07}$ | $86.81_{\pm 0.08}$ | $79.22_{\pm 2.03}$ | $58.49_{\pm 0.06}$ | $57.63_{\pm 0.86}$ | $56.05_{\pm 0.52}$ | $34.71_{\pm 1.90}$ | $32.00_{\pm 1.56}$ | $26.84_{\pm 1.52}$ |
| | DaFKD-hetero | Local | $83.83_{\pm 1.01}$ | $77.69_{\pm 2.06}$ | $64.71_{\pm 2.29}$ | $52.76_{\pm 3.33}$ | $48.99_{\pm 1.73}$ | $39.98_{\pm 5.09}$ | $33.85_{\pm 0.96}$ | $26.02_{\pm 3.19}$ | $22.26_{\pm 4.49}$ |
| | | Global | $89.26_{\pm 0.31}$ | $87.84_{\pm 0.64}$ | $80.07_{\pm 1.23}$ | $58.97_{\pm 0.43}$ | $58.52_{\pm 0.19}$ | $57.33_{\pm 1.00}$ | $35.69_{\pm 1.48}$ | $32.57_{\pm 1.34}$ | $26.54_{\pm 1.73}$ |
| Hetero-FL | FedMD | Local | $70.33_{\pm 3.98}$ | $63.47_{\pm 4.13}$ | $60.67_{\pm 5.09}$ | $42.28_{\pm 6.11}$ | $39.64_{\pm 5.45}$ | $26.90_{\pm 7.70}$ | $23.84_{\pm 3.76}$ | $19.37_{\pm 3.20}$ | $13.89_{\pm 6.70}$ |
| | | Global | $78.91_{\pm 2.32}$ | $75.48_{\pm 3.31}$ | $66.67_{\pm 2.50}$ | $46.37_{\pm 3.11}$ | $49.40_{\pm 4.65}$ | $39.44_{\pm 9.89}$ | $26.37_{\pm 2.80}$ | $22.60_{\pm 2.43}$ | $19.42_{\pm 3.17}$ |
| | MSFKD | Local | $82.26_{\pm 0.63}$ | $77.31_{\pm 3.04}$ | $62.52_{\pm 9.36}$ | $52.06_{\pm 2.20}$ | $49.94_{\pm 1.27}$ | $39.72_{\pm 3.70}$ | $34.30_{\pm 3.11}$ | $27.93_{\pm 2.71}$ | $22.59_{\pm 5.84}$ |
| | | Global | $87.79_{\pm 0.18}$ | $86.98_{\pm 0.31}$ | $80.00_{\pm 1.75}$ | $58.88_{\pm 0.38}$ | $59.09_{\pm 0.24}$ | $56.29_{\pm 0.83}$ | $36.29_{\pm 0.60}$ | $31.62_{\pm 0.87}$ | $26.70_{\pm 2.33}$ |
| | FedGD | Local | $82.40_{\pm 1.01}$ | $76.02_{\pm 1.39}$ | $63.71_{\pm 8.06}$ | $51.02_{\pm 2.07}$ | $50.17_{\pm 1.86}$ | $39.17_{\pm 2.78}$ | $35.08_{\pm 2.55}$ | $26.50_{\pm 1.64}$ | $23.47_{\pm 4.83}$ |
| | | Global | $87.52_{\pm 0.24}$ | $87.22_{\pm 0.13}$ | $79.31_{\pm 0.75}$ | $59.26_{\pm 0.41}$ | $58.03_{\pm 0.26}$ | $56.34_{\pm 0.65}$ | $36.86_{\pm 1.06}$ | $30.66_{\pm 1.59}$ | $27.53_{\pm 2.20}$ |
| | **FedFD (ours)** | Local | $\mathbf{84.91_{\pm 0.42}}$ | $\mathbf{78.03_{\pm 1.49}}$ | $\mathbf{65.33_{\pm 9.22}}$ | $\mathbf{54.98_{\pm 1.76}}$ | $\mathbf{52.24_{\pm 1.90}}$ | $\mathbf{41.68_{\pm 4.12}}$ | $\mathbf{36.78_{\pm 5.13}}$ | $\mathbf{30.90_{\pm 5.25}}$ | $\mathbf{23.41_{\pm 4.99}}$ |
| | | Global | $\mathbf{90.06_{\pm 0.03}}$ | $\mathbf{89.64_{\pm 0.23}}$ | $\mathbf{82.74_{\pm 0.58}}$ | $\mathbf{61.07_{\pm 0.22}}$ | $\mathbf{60.86_{\pm 0.10}}$ | $\mathbf{59.24_{\pm 0.48}}$ | $\mathbf{40.27_{\pm 1.33}}$ | $\mathbf{34.24_{\pm 1.13}}$ | $\mathbf{29.09_{\pm 1.69}}$ |

method with the following baselines. **(1) Representative FL models:** HeteroFL [6], MOON-hetero [22]; **(2) FL for homogeneous model:** FedFusion-hetero [33], FedGen-hetero [58], DaFKD-hetero [49]; **(3) FL for heterogeneous model:** FedMD [21], MSFKD [48], FedGD [55].

**Configurations:** Unless otherwise mentioned, we set the number of local training epoch $E = 10$, communication round $T = 200$, and the client number $K = 20$ with an active ratio $r = 0.4$. For local training, the batch size is 64 and the weight decay is $1e - 4$. The learning rate is 0.01 for distillation and 0.001 for training the local model. For the model on the server, we employ ResNet-18 [11] as the basic backbone. Like [6], we construct ten different computation complexity levels $\{a, b, c, \dots j\}$ with the hidden channel decay rate 10%. For example, model "a" has all the model parameters, while models "b" and "c" have the 90% and 80% effective parameters. We use "a-d-g" three different model architectures in our experiments and conduct more experiments with other architectures in Section 4.2. Similarly to [6], we have conducted statistics based on the performance of the global model for all secondary experiments.

## 4.2 Performance Overview

**Test Accuracy.** Table 1 shows the test accuracy of various methods with heterogeneous data across three datasets. Each experiment set is run twice, and we take each run's final ten rounds' accuracy and calculate the average value and standard variance. We report both the global model accuracy and the average accuracy of all local models. Firstly, we observe that the performance of the global model significantly outpaces that of the local model. For the Classic FL method, MOON does not achieve a substantial advantage with heterogeneous models, suggesting that the contrastive loss between different model architectures provides limited improvement to the method. As for the Homo-FD method, while logit distillation introduces performance fluctuations, it offers some enhancement after an increase in training epochs. Both FedGen and DaFKD are data-free distillation methods, and compared to FedFusion, their performance improvements are not as significant as reported in the original papers. A possible reason is that the training of the generator remains unstable, especially considering we employ relatively complex datasets instead of handwriting-digit recognition datasets like MNIST. Regarding Hetero-FL methods, FedMD performs poorly due to the absence of aggregated parameters. The other two methods follow the logit distillation paradigm and exhibit superior performance. FedFD achieves the best performance in all cases by up to 16.09% in terms of global model accuracy.

**Communication Efficiency.** Figure.2 (Table) compares the communication efficiency of various methods by measuring the communication rounds required to achieve the test accuracy and Figure.4 focuses on the learning curves. Although knowledge distillation methods can accelerate model convergence, the training process of logit distillation is unstable, as we have detailed above. With the orthogonal projection technique in feature distillation, FedFD achieves the fastest convergence with a stable training process.

**Ablation Study.** As shown in Table 3, we evaluate the effects of each module in our model via ablation studies. -w/o feature alignment and -w/o orthogonal projection denote the performance of our model without using hierarchical feature alignment and orthogonal projection. Compared with FedFD, the performance of FedFD -w/o orthogonal projection and FedFD -w/o orthogonal projection degrades evidently with a range of 0.63%~2.43%. While moving both components, the performance will drop significantly. Specifically, the orthogonal projection technique plays a much more important role with the obvious improvement in test accuracy,

Table 2: Evaluation of different baselines on three datasets ($\alpha = 1.0$), in terms of the number of communication rounds to reach target test accuracy (acc). The best results are **bold**.

| Methods | CIFAR-10 | | CIFAR-100 | | Tiny-ImageNet | |
|---|---|---|---|---|---|---|
| | acc=70% | acc=80% | acc=55% | acc=60% | acc=25% | acc=30% |
| HeteroFL | $12_{\pm3}$ | $25_{\pm6}$ | $20_{\pm3}$ | $>200$ | $124_{\pm16}$ | $>200$ |
| MOON-hetero | $11_{\pm4}$ | $26_{\pm7}$ | $22_{\pm9}$ | $>200$ | $100_{\pm9}$ | $188_{\pm7}$ |
| FedFusion-hetero | $20_{\pm7}$ | $52_{\pm7}$ | $61_{\pm11}$ | $>200$ | $96_{\pm25}$ | $153_{\pm28}$ |
| FedGen-hetero | $21_{\pm11}$ | $38_{\pm15}$ | $198_{\pm20}$ | $>200$ | $115_{\pm12}$ | $176_{\pm24}$ |
| DaFKD-hetero | $23_{\pm9}$ | $41_{\pm9}$ | $44_{\pm13}$ | $191_{\pm27}$ | $85_{\pm8}$ | $126_{\pm21}$ |
| FedMD | $62_{\pm20}$ | $>200$ | $139_{\pm21}$ | $>200$ | $>200$ | $>200$ |
| MSFKD | $19_{\pm9}$ | $45_{\pm17}$ | $33_{\pm10}$ | $171_{\pm20}$ | $87_{\pm22}$ | $115_{\pm31}$ |
| FedGD | $17_{\pm8}$ | $34_{\pm5}$ | $40_{\pm12}$ | $189_{\pm21}$ | $106_{\pm14}$ | $150_{\pm23}$ |
| **FedFD (ours)** | $\mathbf{10_{\pm5}}$ | $\mathbf{20_{\pm9}}$ | $\mathbf{16_{\pm6}}$ | $\mathbf{60_{\pm18}}$ | $\mathbf{67_{\pm8}}$ | $\mathbf{99_{\pm19}}$ |

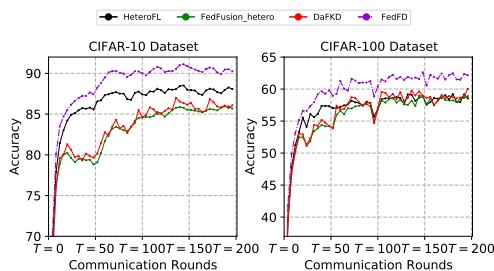

Figure 4: Convergence and efficiency comparison of various methods on two datasets.

and the hierarchical feature alignment technique has a relatively small impact on test accuracy, which may be due to the scale of the experiment, but it can save storage costs. Experiment results verify that all modules are essential to train a robust Hetero-FL model.

Table 3: Ablation study of FedFD with two main components.

| Method | CIFAR-10 | | CIFAR-100 | |
|---|---|---|---|---|
| | $\alpha$=1.0 | $\alpha$=0.1 | $\alpha$=1.0 | $\alpha$=0.1 |
| FedFD | **89.64** | **82.74** | **60.86** | **59.24** |
| -w/o feature alignment | 89.01 | 81.56 | 59.77 | 58.67 |
| -w/o orthogonal projection | 87.96 | 80.92 | 59.33 | 57.29 |
| -w/o both components | 85.70 | 78.47 | 58.86 | 56.12 |

Table 4: Evaluation of combination of various client-side models levels for CIFAR-10 dataset ($\alpha = 1.0$).

| Method | Model Heterogeneity | | | |
|---|---|---|---|---|
| | a-d-g | a-f | b-d-f | a-c-d-f-i |
| HeteroFL | $87.53_{\pm0.15}$ | $88.15_{\pm0.18}$ | $85.01_{\pm0.30}$ | $82.53_{\pm0.27}$ |
| FedFusion-hetero | $85.70_{\pm0.15}$ | $86.85_{\pm0.10}$ | $85.37_{\pm0.28}$ | $84.10_{\pm0.15}$ |
| FedGD | $87.96_{\pm0.08}$ | $88.47_{\pm0.21}$ | $85.23_{\pm0.40}$ | $84.01_{\pm0.46}$ |
| FedFD | $\mathbf{89.64_{\pm0.23}}$ | $\mathbf{89.92_{\pm0.19}}$ | $\mathbf{87.47_{\pm0.35}}$ | $\mathbf{85.92_{\pm0.28}}$ |

**Data Heterogeneity.** Table 1 illustrates the variation in test accuracy across different levels of data heterogeneity. A clear trend emerges, with all methods exhibiting improved accuracy as data heterogeneity decreases. Simultaneously, we observed that the data heterogeneity has a greater impact on local models than on global models. Specifically, when $\alpha = 0.1$, the performance of local models across all baselines on CIFAR-100 declines significantly, this underscores the urgent need to investigate Hetero-FL, as they are more susceptible to the influence of data distribution. Notably, FedFD consistently outperforms other methods, achieving the most significant improvements across all settings.

**Parameter Sensitivity Analysis.** In this section, we first explore the model heterogeneity issue with different models. As shown in Table 4, we conduct experiments using four different models and define that a larger number of models represented a higher level of model heterogeneity. As the degree of heterogeneity increased, the performance of all methods declined, with FedFusion showing a particularly significant drop. This verified the shortcoming of logit distillation in the context of model heterogeneity. In contrast, FedFD consistently retains superior performance.

Table 5: The scalability of FedFD and other baselines ($\alpha$=0.01).

| Dataset | HeteroFL | FedFusion-hetero | MSFKD | FedFD |
|---|---|---|---|---|
| CIFAR-10 | $63.29_{\pm5.19}$ | $64.14_{\pm8.16}$ | $65.30_{\pm6.77}$ | $\mathbf{67.03_{\pm9.34}}$ |
| CIFAR-100 | $44.59_{\pm1.32}$ | $45.31_{\pm1.09}$ | $44.90_{\pm2.73}$ | $\mathbf{48.05_{\pm2.41}}$ |
| Tiny-ImageNet | $21.86_{\pm4.33}$ | $22.97_{\pm2.89}$ | $23.26_{\pm2.54}$ | $\mathbf{25.12_{\pm1.53}}$ |

Table 6: The different architectures of models (CNN and ResNet).

| Method | CIFAR-10 | | CIFAR-100 | | Tiny-ImageNet | |
|---|---|---|---|---|---|---|
| | $\alpha$=1.0 | $\alpha$=0.1 | $\alpha$=1.0 | $\alpha$=0.1 | $\alpha$=10.0 | $\alpha$=1.0 |
| FedMD | $67.59_{\pm3.61}$ | $53.10_{\pm9.23}$ | $30.52_{\pm3.98}$ | $24.47_{\pm1.70}$ | $21.29_{\pm1.67}$ | $19.95_{\pm2.80}$ |
| FedFD | $\mathbf{71.28_{\pm1.25}}$ | $\mathbf{59.51_{\pm7.13}}$ | $\mathbf{34.93_{\pm0.26}}$ | $\mathbf{28.98_{\pm0.74}}$ | $\mathbf{30.79_{\pm1.28}}$ | $\mathbf{27.96_{\pm0.47}}$ |

Then, we examine the scalability of our method. As shown in Table 5, although all methods exhibit significant performance degradation in large-scale experiments, forcing some clients to own very few samples, FedFD outperforms other baselines, demonstrating its robustness and effectiveness.

**Model Architecture.** Although the above experiments validate the effectiveness of FedFD within the HeteroFL framework, *FedFD fundamentally constitutes an advanced feature distillation method independent of model architectures and parameter aggregation strategies*. We selected HeteroFL as the framework due to its widespread recognition, which facilitates comprehensive comparisons with other baselines. In Table 6, we conducted experiments using two architectures to mitigate potential impacts arising from the HeteroFL framework. Here, we adopt the FedMD to perform knowledge distillation to fuse client knowledge directly. The experimental results demonstrate the adaptability of FedFD with diverse architectures.

# 5    Conclusion

We introduced FedFD, a straightforward framework, to tackle feature distillation using heterogeneous models within federated learning. FedFD serves as a minimal personalization extension for any federated learning algorithms involving global model aggregation, ensuring privacy and communication efficiency. Comprehensive experiments demonstrate that FedFD can enhance test accuracy.

## Acknowledgments and Disclosure of Funding

This work is supported by the National Key Research and Development Program of China under grant 2024YFC3307900; the National Natural Science Foundation of China under grants 62376103, 62302184, 62436003 and 62206102; Major Science and Technology Project of Hubei Province under grant 2024BAA008; Hubei Science and Technology Talent Service Project under grant 2024DJC078; Ant Group through CCF-Ant Research Fund; and Fundamental Research Funds for the Central Universities under grant YCJJ20252319. The computation is completed in the HPC Platform of Huazhong University of Science and Technology.

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
