# OpenReview forum: "Feature Distillation is the Better Choice for Model-Heterogeneous Federated Learning"
_NeurIPS.cc/2025/Conference — NeurIPS 2025 poster_

### Official Review · Reviewer_bfr5 · 2025-06-30

**Clarity:** 3
**Significance:** 3
**Originality:** 4
**Rating:** 5
**Confidence:** 5

**Summary:**

This paper introduces a feature distillation method to enhance model-heterogeneous federated learning by addressing the limitations of traditional logit distillation. FedFD employs orthogonal projection to align feature representations from diverse client-side models. The proposed method is evaluated on multiple datasets, demonstrating superior performance compared to state-of-the-art baselines.

**Questions:**

How does FedFD perform in real-world applications, such as healthcare or autonomous driving, where data distributions and model architectures can be highly heterogeneous?

**Ethical Concerns:**

["NO or VERY MINOR ethics concerns only"]

**Final Justification:**

The author has addressed all my concerns. I would support this paper

**Limitations:**

Yes

**Quality:**

3

**Strengths And Weaknesses:**

Strengths:

1. This paper is well-written and easy to follow with open-source code.

2. The paper proposes a novel feature distillation method, which can effectively addresses the challenge of aggregating knowledge from heterogeneous models in FL.

3. The use of orthogonal projection to align feature representations is a unique and theoretically sound technique.

4. The authors conduct extensive experiments on multiple datasets and various model architectures, providing strong empirical evidence for the effectiveness of FedFD.

Major Weaknesses:

1. Although FedFD maintains privacy by not sharing raw data, the use of feature representations and projection layers could introduce new privacy risks that are not thoroughly discussed.

2. The introduction of orthogonal projection layers and hierarchical feature alignment may increase the complexity of implementation and training, potentially limiting practical deployment.

Minor Weaknesses:

1. The scalability of FedFD to extremely large models, such as those with billions of parameters, is not discussed, which could be a limitation in practical applications.

2. The paper could benefit from a more detailed comparison with data-free distillation methods, which are gaining traction in federated learning due to their ability to work without access to client data.

3. The assumption that feature dimensions can be aligned using orthogonal projections may not hold in all cases, especially when client-side models have vastly different architectures.

---

> ### Author Rebuttal · Authors · 2025-07-29
>
> Thank you for providing such valuable comments. We have carefully addressed these critical comments, and responses have been given one by one.
>
> > **W1. Concerns about potential privacy risk by using two modules in FedFD.**
>
> **R1:** Thank you for this valuable comment. We would like to clarify that the **projection layers** in FedFD are deployed and stored entirely on the **server side**, and do not require transmission between clients and the server. As a result, they do not introduce any additional privacy risks. Regarding the **feature representations** used for distillation, it is important to note that, similar to traditional logit-based distillation, information such as logits also needs to be transmitted. Both logits and features reflect the model’s behavior on the **public dataset**, which is assumed to carry **no privacy risks**, as it contains no sensitive or user-specific data. Moreover, these features and logits are generated by running the distillation data through client models, which are already uploaded to the server for parameter aggregation as part of standard federated learning procedures. Therefore, **FedFD does not introduce any additional privacy leakage risks compared to FedAvg**.
>
> > **W2. Concerns about training cost of two modules in FedFD.**
>
> **R2:** Thank you for raising this point. We would like to clarify that the two modules introduced in FedFD will not limit the practical deployment. First, the **hierarchical feature alignment** module is essentially a feature aggregation strategy that does not introduce additional trainable parameters. Although it aggregates multiple feature representations, these are stored and processed on the **server side**, not the clients. In the context of federated learning, it is standard practice to focus on client-side overhead, as servers are generally assumed to have sufficient computational and storage capacity. Second, while the **orthogonal projection** module does introduce new parameters, these computations are also conducted **entirely on the server**. Therefore, this module does not increase the resource requirements for client devices. Overall, both modules are designed to keep the client-side lightweight, and thus do not hinder deployment on edge devices.
>
> > **W3. Concerns about scalability of large models.**
>
> **R3:** Thank you for this insightful comment. Deploying models with billions of parameters pose scalability challenges, especially given the resource constraints of edge devices in federated learning scenarios. Addressing such large-scale models is beyond the primary scope of our current work. There is already a growing body of research specifically focusing on combining federated learning with LLMs, which tackles these challenges in depth. Due to the limited rebuttal time and the high computational cost involved, we are unable to conduct large-scale experiments involving billion-parameter models at this stage. We acknowledge the importance of this direction and will consider extending our framework to support such models in the final version of the paper, as well as exploring it more thoroughly as part of our future work.
>
> > **W4. Concerns about generated proxy data.**
>
> **R4:** Thank you for this valuable comment. To validate this, we conduct additional experiments where we employ the three data-free methods to synthesize distillation data and evaluate the effectiveness of FedFD. Unlike logit-based distillation, even though the quality of synthetic data may not match that of assumed high-quality public datasets, FedFD leverages feature-level outputs to capture richer knowledge, resulting in no significant performance degradation.
>
> | Methods          | Metrics | CIFAR-10 ($\alpha=1.0$) | CIFAR-100 ($\alpha=1.0$) |
> | ---------------- | ------- | ----------------------- | ------------------------ |
> | **FedFD-Public** | Local   | 78.03                   | 52.24                    |
> |                  | Global  | 89.64                   | 60.86                    |
> | FedFD-FedGen     | Local   | 77.12                   | 50.93                    |
> |                  | Global  | 87.99                   | 58.91                    |
> | FedFD-FedFTG     | Local   | 77.04                   | 51.35                    |
> |                  | Global  | 88.11                   | 59.14                    |
> | FedFD-DaFKD      | Local   | 77.58                   | 51.45                    |
> |                  | Global  | 88.97                   | 60.02                    |
>
> Furthermore, to ensure fair comparison in our experiments, the public dataset used for distillation is deliberately chosen to have distributional differences from the training dataset. Specifically, we use {CIFAR-100, Tiny-ImageNet, ImageNet} as the distillation datasets for {CIFAR-10, CIFAR-100, Tiny-ImageNet}.
>
> > **W5. Concerns about model architectures.**
>
> **R5:** Thank you for your careful review. In Table 6 of our original paper, we have already conducted experiments to address the reviewer's concern regarding models of varying architecture (e.g., ResNet, CNN). These experiments evaluate the performance of our method alongside the model-agnostic approach FedMD, demonstrating the superiority of feature-based distillation under heterogeneous model settings.
>
> Nevertheless, we primarily adopt the sub-model partitioning approach because it reflects a common and practical scenario in real-world federated learning deployments. For example, **in UAV clusters or joint ventures between subsidiaries with a trusted terminal—where they often follow their own protocol standards that define the set of model architectures to be used, rather than allowing each client to independently choose entirely different model architectures.** Therefore, there is no need to deploy entirely different model architectures that would introduce significant bias and training overhead. In contrast, adapting model based on hardware capabilities is more practical and cost-efficient.
>
> > **Q1. Real-world applications suitable for FedFD.**
>
> **A1:** Thanks for this question. As we mentioned in **R5**, UAV clusters represent a realistic and well-suited deployment scenario for FedFD. In UAV-based sensing tasks, lightweight models are typically required for on-device deployment, and due to standardized protocols, it is uncommon to deploy models with drastically different architectures across devices. This makes the sub-model partitioning setting particularly applicable. Moreover, data collected in different deployment scenarios often exhibit a certain degree of heterogeneity, while public data can be readily collected on the server side. This combination makes UAV-based applications a representative and practical use case for heterogeneous federated learning.

---

> > ### Comment · Reviewer_bfr5 · 2025-08-04
> >
> > Thank you for the authors' reply. Most of my concerns have been addressed, and after reading the other reviewers' comments, I would like to  maintain my scores.  I think most of the issues mentioned by other reviewers are related and may not undermine the main contributions of the paper. I would like to support this paper.

---

### Official Review · Reviewer_AyEq · 2025-07-01

**Clarity:** 2
**Significance:** 2
**Originality:** 1
**Rating:** 3
**Confidence:** 3

**Summary:**

The proposed framework FedFD focuses on federated learning for edge clients with heterogeneous model architectures. To address the limitations of traditional logits aggregation, FedFD utilizes orthogonal projection to ensure linear transformations of features, preventing knowledge conflicts and preserving feature shapes.

**Questions:**

Please refer to the Weaknesses section for specific limitations and suggestions for improvement.

**Ethical Concerns:**

["NO or VERY MINOR ethics concerns only"]

**Final Justification:**

My concerns are mostly addressed. I will raise my original score. But I still concern about the contribution of this work.

**Limitations:**

Yes.

**Paper Formatting Concerns:**

None.

**Quality:**

3

**Strengths And Weaknesses:**

Strengths:

1.The proposed feature distillation technique, FedFD, tackles the challenges of model heterogeneity in federated learning by using orthogonal projection to align feature representations.

2.FedFD is designed to be easily integrated into existing federated learning systems.

3.The experiments are solid.

Weaknesses:

1.A potential weakness is that the HeteroFL setup used in this work is not a typical distillation-based heterogeneous federated learning configuration. This setup leans more towards federated learning based on sub-model partitioning, making it difficult to distinguish whether the final results stem from distillation or model aggregation. The authors discuss this issue in Tab. 6, but a more detailed explanation is still needed for this case study.

2.The author's core contribution to this paper, distillation using orthogonal projections, requires a more detailed explanation to enhance readability.

3.The distillation method proposed by the authors bears similarities to "VKD: Improving Knowledge Distillation Using Orthogonal Projections." VKD also focuses on the computational overhead issues of the Cayley transformation and Gram-Schmidt method and proposes an improved feature distillation approach. The authors need to further clarify the differences between their proposed distillation method and VKD. Additionally, the authors should provide a clearer explanation of why their proposed distillation method is more suitable for distributed environments compared to VKD.

---

> ### Author Rebuttal · Authors · 2025-07-29
>
> Thank you very much for this professional review. The critical comments have been addressed carefully, and responses have been given one by one.
>
> > **W1. Concerns about model heterogeneity setting.**
>
> **R1:** Thank you very much for this helpful comment. Regarding the difference between FedFD and aggregation-based methods, **federated knowledge distillation can be viewed as an add-on component to parameter aggregation methods, designed to further improve model performance.** Methods like HeteroFL just serve as base parameter aggregation frameworks, while FedFD leverages a public dataset to enhance the performance of the aggregated models. And for logit-based federated distillation methods, they also enhance the global model aggregated in a FedAvg manner. **Therefore, the performance gap between the distillation-based methods and HeteroFL can be regarded as the performance contribution of the federated distillation technique.** In our experimental setup, we choose HeteroFL as the backbone paradigm for two main reasons:
>
> (1) The channel heterogeneity addressed by HeteroFL is itself a form of model heterogeneity. In this setting, different devices deploy sub-models with similar architectures but varying sizes and computational requirements. Such scenarios are common in practice—for example, **in UAV clusters or joint ventures between subsidiaries with a trusted terminal—where they often follow their own protocol standards that define the set of model architectures to be used, rather than allowing each client to independently choose entirely different model architectures.** Therefore, there is no need to deploy entirely different model architectures that would introduce significant bias and training overhead. In contrast, adapting model based on hardware capabilities is more practical and cost-efficient.
>
> (2) As the first work to explore feature-based federated distillation, we aim to rigorously validate the effectiveness of our method. To do so, it is important to compare against existing state-of-the-art logit-based federated distillation approaches, which typically rely on parameter aggregation to ensure proper convergence. Without such aggregation, these methods tend to struggle since distillation is insufficient to effectively guide the global model aggregation in the early stages of training, it may lead to unstable convergence or even prevent the model from converging properly, resulting in suboptimal performance.. **Therefore, to ensure a fair comparison, we adopt HeteroFL as the backbone paradigm for model aggregation in our experiments.**
>
> We think that referring to **sub-model partitioning** as a form of **model-heterogeneity** FL is reasonable, as the local models on different clients are indeed different. Currently, most works that adopt the HeteroFL paradigm also use titles referring to it as model-heterogeneous federated learning. However, when the architectural differences between local models are extremely large—for example, completely different networks such as ResNet and CNN—it may be more appropriate to categorize such a setting as a **model-agnostic** FL paradigm. This conceptual distinction currently exists in the field of federated learning but has not yet been strictly defined or clearly separated.
>
> > **W2. Detailed explanation of orthogonal projection technique.**
>
> **R2:** Thank you for this valuable comment. The core contribution of the paper is to firsts introduce a novel feature-based distillation framework, FedFD, for model-heterogeneous federated learning. Instead of relying on traditional logit distillation, which fails to capture the internal representation differences across diverse client models, FedFD distills knowledge at the feature level. It first groups client models by architecture and aggregates their feature representations into clusters. For each cluster, the server maintains a projection layer to align the global model’s features with the client-derived ones. This hierarchical feature alignment enables the server to extract structurally coherent knowledge from clients with varying architectures, enhancing generalization without requiring structural uniformity.
>
> A key innovation in FedFD is the use of orthogonal projections to construct these projection layers. **To ensure effective and conflict-free alignment of feature spaces**, we use the skew-symmetric matrix $\textbf{W}_d$ to obtain the orthogonal project layer parameters $\mathcal{M}_d$ by truncating the column vectors of exp($\textbf{W}_d$). Firstly, the parameters of $\text{W}_d$ are randomly initialized. Then, the client updates $\textbf{W}_d$ through back-propagation with Eq.(9). The exponential map of this matrix $\exp(\textbf{W}_d)$ produces an orthogonal matrix with desirable mathematical properties, such as preserving vector norms and ensuring rotational transformation without distortion. **Because feature dimensions across clients may differ**, only a subset of columns from $\exp(\textbf{W}_d)$ is retained to match the dimensionality of the target space. **This approach ensures each projection layer maps global model features into independent, non-overlapping subspaces, thus avoiding feature conflicts.**
>
> The server then updates both the global model and the projection layers by minimizing KL divergence between the projected global features and the aggregated client features. This orthogonal projection mechanism is efficient, scalable, and leads to more stable and accurate model training.
>
> > **W3. Differences and specific contributions compared to VKD.**
>
> **R3:** Thank you for raising this concern. We cited VKD [28] in our paper as a related work but did not conduct an in-depth comparison, as the two approaches differ significantly in terms of task setting, technical motivation, and methodological details. Specifically, orthogonal projection is a commonly used technique across various domains, including federated continual learning and machine unlearning. Methods such as the Cayley transform and Gram-Schmidt method are well-studied and are not novel problems now. **Neither VKD nor FedFD aims to innovate on the computational efficiency of these projection methods, as this has already been extensively studied in prior knowledge distillation and mathematical literature.** In FedFD, we simply adopt a computationally convenient orthogonalization technique from the perspective of federated learning to achieve parameter orthogonalization and accelerate the experiments. **Notably, the computational formulations in FedFD and VKD are also different.**
>
> Furthermore, **VKD focuses on preserving intra-batch feature similarity** through orthogonal projection. It analyzes how orthogonal projection can better preserve similar features and further enhances feature quality using standardized normalization. In contrast, FedFD addresses a different challenge: under model heterogeneity, each client should contribute as a teacher model to distill the global model. In knowledge distillation, a larger number of distillation samples typically leads to better effectiveness, as they better represent the knowledge embedded in the training data. This can result in a full-rank knowledge matrix, making the projected knowledge also full-rank, which behaves like a non-linear mapping. Consequently, the global model may fail to properly interpret the knowledge, leading to conflicts. To address this, **FedFD is designed to use orthogonal projection to map knowledge from different projection layers into orthogonal subspaces, thereby preventing knowledge conflict and maximizing knowledge transfer.** **FedFD does not include a normalization module, as normalization is not the primary challenge our work aims to address.**
>
> Lastly, in terms of technical implementation: each projection in FedFD is **column-orthonormal** to avoid knowledge conflict, whereas VKD uses **row-orthonormal** projections for orthogonal reparameterization. VKD also users the logit distillation to enhance the performance which is not used in FedFD. FedFD focuses on federated distillation under model heterogeneity, which is a critical yet underexplored direction in distributed learning. Rather than solely aiming for performance improvement as in traditional ML algorithm research, **our work identifies specific challenges brought by heterogeneous models and proposes a novel solution that combines hierarchical feature aggregation and orthogonal projection.** These components not only address practical issues in knowledge alignment and conflict but also contribute methodological innovations to the field.
>
> **Why suitable:** VKD is designed for centralized knowledge distillation involving a single teacher model and a single student model. It cannot be directly applied to distributed settings, where the teacher models are constructed by aggregating outputs from multiple client models. Moreover, VKD does not address the problem of feature knowledge conflicts among different teacher models, which is a key challenge in federated settings. In summary, VKD does not consider the specific challenges present in distributed learning scenarios, and there is no clear motivation to adopt VKD in such contexts. VKD and FedFD focus on fundamentally different problems, with only superficial methodological similarities but entirely different motivations and application goals.
>
> **PS: We have carefully addressed the concerns you raised in our rebuttal. If there are any remaining questions or clarifications needed, please feel free to point them out during the discussion phase. We sincerely hope that our work will receive your recognition following the rebuttal and discussion. We are looking forward to hearing your feedback soon!**

---

> > ### Author Response · Authors · 2025-08-05
> >
> > Dear reviewer AyEq, during this rebuttal period, we have:
> >
> > - Explain the model heterogeneity setting
> > - Provide a detailed description of the orthogonal projection technique
> > - Explain differences and specific contributions compared to VKD.
> >
> > In this rebuttal, all the main concerns about heterogeneity that the esteemed reviewers raised are similar, **and our responses have received acknowledgment and recognition from the other two reviewers**. We would love to hear your feedback on our updates and look forward to discussing any remaining concerns you may have. Thank you for your time and consideration.

---

> > > ### Comment · Reviewer_AyEq · 2025-08-07
> > >
> > > Thank you for your detailed response. My concerns are mostly addressed. I will raise my original score.

---

> > > > ### Author Response · Authors · 2025-08-07
> > > >
> > > > Thanks for your timely reply! Your professional suggestions have greatly improved our work.
> > > >
> > > > We understand that your initial rating was negative, and we realize that a single-point increase might not change the paper's final decision. We hope you can reconsider our score based on the reviewers' responses. Please feel free to reach out to us if you have any further questions. **Your recognition and positive adjustment mean a great deal to us.**
> > > >
> > > > Thank you for your valuable suggestions and professional review.
> > > >
> > > > Best wishes,

---

### Official Review · Reviewer_gifN · 2025-07-02

**Clarity:** 4
**Significance:** 3
**Originality:** 3
**Rating:** 5
**Confidence:** 5

**Summary:**

This paper proposes FedFD, a feature distillation framework for model-heterogeneous federated learning with feature alignment and orthogonal projection to mitigate knowledge bias across clients. Evaluated on CIFAR-10/100 and Tiny-ImageNet under varying data heterogeneity, FedFD outperforms all SOTA methods and demonstrates faster convergence and robustness.

**Questions:**

1. For data-free deployment, how would FedFD integrate with generator-based methods? Would feature distillation require adjustments?

2. In Figure 3, logit distillation destabilizes Hetero-FL training. Does feature distillation alone explain FedFD’s stability, or is orthogonal projection the key factor?

**Ethical Concerns:**

["NO or VERY MINOR ethics concerns only"]

**Final Justification:**

Although the paper has certain limitations in its setting, it still demonstrates considerable potential for broad applicability. Since all raised concerns have been properly addressed, I decide to keep my score.

**Limitations:**

yes

**Paper Formatting Concerns:**

NA.

**Quality:**

3

**Strengths And Weaknesses:**

Strengths:

1. The paper is well-written and easy to follow.

2. The presentation is quite good and the mathematical formulation is rigorous.

3. The proposed method is a technically sound solution to mitigate knowledge bias across heterogeneous models.

4. Extensive experiments have been done to verify the effectiveness in various settings.

Weaknesses:

1. FedFD relies on a proxy dataset like prior FD works. While mentioned as flexible, experiments only use additional real data but no tests with generated data.

2. Hetero-FL baselines are logit-based. Comparisons to feature-distillation FL methods or personalized FL would strengthen claims.

3. Maintaining projection layers per client architecture could increase server storage or computation budget. I suggest that authors analyze and propose a solution to this issue.

---

> ### Author Rebuttal · Authors · 2025-07-29
>
> We thank you very much for providing the positive comments. In the following, we give detailed responses to each review.
>
> > **W1&Q1. Concerns about generated proxy data.**
>
> **R1:** Thanks a lot for raising this concern. **Federated knowledge distillation can be viewed as an add-on component to parameter aggregation methods, designed to further improve model performance.** HeteroFL serve as base parameter aggregation frameworks, while FedFD leverages a public dataset to enhance the performance of the aggregated models. And for logit-based federated distillation methods, they also enhance the global model aggregated in a FedAvg manner. To validate this, we conduct additional experiments where we employ the three data-free methods to synthesize distillation data and evaluate the effectiveness of FedFD. Unlike logit-based distillation, even though the quality of synthetic data may not match that of assumed high-quality public datasets, FedFD leverages feature-level outputs to capture richer knowledge, resulting in no significant performance degradation.
>
> | Methods          | Metrics | CIFAR-10 ($\alpha=1.0$) | CIFAR-100 ($\alpha=1.0$) |
> | ---------------- | ------- | ----------------------- | ------------------------ |
> | **FedFD-Public** | Local   | 78.03                   | 52.24                    |
> |                  | Global  | 89.64                   | 60.86                    |
> | FedFD-FedGen     | Local   | 77.12                   | 50.93                    |
> |                  | Global  | 87.99                   | 58.91                    |
> | FedFD-FedFTG     | Local   | 77.04                   | 51.35                    |
> |                  | Global  | 88.11                   | 59.14                    |
> | FedFD-DaFKD      | Local   | 77.58                   | 51.45                    |
> |                  | Global  | 88.97                   | 60.02                    |
>
> Furthermore, to ensure fair comparison in our experiments, the public dataset used for distillation is deliberately chosen to have distributional differences from the training dataset. Specifically, we use {CIFAR-100, Tiny-ImageNet, ImageNet} as the distillation datasets for {CIFAR-10, CIFAR-100, Tiny-ImageNet}.
>
> > **W2. Concerns about other feature-distillation or personalized baselines.**
>
> **R2:** Thank you for this helpful comment. We apologize if this was not sufficiently clear. To the best of our knowledge, **we are the first to study feature-based distillation in federated learning**. Therefore, our baselines are selected from two perspectives: state-of-the-art methods in model-heterogeneous federated learning and logit-based federated distillation. As for traditional FL methods, we have included MOON under the HeteroFL paradigm for comparison. Overall, our choice of baselines and datasets is consistent with or even more advanced than those used in existing related works.
>
> > **W3. Concerns about storage or computational cost for projection layers.**
>
> **R3:**  Thank you for your valuable comment. We would like to clarify that the two modules introduced in FedFD will not limit the practical deployment. First, the **hierarchical feature alignment** module is essentially a feature aggregation strategy that does not introduce additional trainable parameters. Although it aggregates multiple feature representations, these are stored and processed on the **server side**, not the clients. In the context of federated learning, it is standard practice to focus on client-side overhead, as servers are generally assumed to have sufficient computational and storage capacity. Second, while the **orthogonal projection** module does introduce new parameters, these computations are also conducted **entirely on the server**. Therefore, this module does not increase the resource requirements for client devices. Overall, both modules are designed to keep the client-side lightweight, and thus do not hinder deployment on edge devices.
>
> > **Q2.Key factor contributing to stable Hetero-FL training.**
>
> **A2:**  Thank you for this question. In Figure 3, the paper demonstrates that logit distillation causes instability in Hetero-FL training, as the aggregation of soft predictions fails to address the misalignment in feature spaces between heterogeneous models. The instability arises because logits reflect only the output layer and are sensitive to variations in model architectures, leading to ineffective and misleading knowledge distillation across clients. FedFD addresses this issue not merely through feature distillation alone, but by combining it with orthogonal projection. The ablation study (Table 3) confirms that removing the orthogonal projection module significantly degrades performance, more so than removing hierarchical feature alignment. T**his suggests that while feature distillation helps align internal representations, the orthogonal projection is the key factor that ensures stability and effectiveness.** It resolves knowledge conflicts by mapping features into orthogonal subspaces, thus maximizing the knowledge transfer and enabling consistent model updates in heterogeneous federated settings.

---

> > ### Comment · Reviewer_gifN · 2025-08-04
> >
> > Thanks for author's rebuttal. Although the paper has certain limitations in its setting, it still demonstrates considerable potential for broad applicability. Since all raised concerns have been properly addressed, I decide to keep my score.

---

> > > ### Author Response · Authors · 2025-08-05
> > >
> > > Thanks for your timely reply. We are glad to know that your concerns have been effectively addressed. Your professional suggestions have greatly improved our work. We truly appreciate your efforts and recognition of our research.
> > >
> > > Best wishes,

---

### Official Review · Reviewer_zr5J · 2025-07-03

**Clarity:** 3
**Significance:** 2
**Originality:** 2
**Rating:** 3
**Confidence:** 4

**Summary:**

This paper proposes a model-heterogeneous FL framework, FedFD, that improves information integration between different models through feature distillation. Compared to existing methods that rely on logit distillation, FedFD effectively compensates for knowledge bias from heterogeneous models. Furthermore, FedFD incorporates aligned feature information via orthogonal projection to maximize the distilled knowledge. Extensive experiments demonstrate that FedFD achieves superior performance compared to state-of-the-art methods.

**Questions:**

1. The paper needs to address the challenge of obtaining high-quality public datasets that correlate well with local data. The authors should elaborate on how this requirement might impact privacy considerations in federated learning settings and propose strategies to mitigate these concerns. Additionally, the authors should discuss whether their method would still be effective in scenarios where no public datasets are available.
2. While the paper aims to address model heterogeneity in FL, the experiments only consider variations in channel numbers rather than truly heterogeneous model architectures (e.g., ResNet vs. CNN vs. MobileNet). The authors should demonstrate how their method would perform with more significant architectural differences. Additionally, for channel heterogeneity specifically, HeteroFL appears to be a more suitable approach. The authors need to clarify the key distinctions between FedFD and existing approaches like HeteroFL and FedRolex[2].
3. The paper would benefit from theoretical analysis supporting the effectiveness of FedFD, including theoretical guarantees or convergence analysis. The experiments are limited to relatively simple datasets like CIFAR, and the authors should consider testing on more complex, real-world datasets to demonstrate broader applicability.
4. The formatting inconsistencies in the paper (e.g., font size variations in lines 270-271) should be addressed in a revised version.

[2] FedRolex: Model-Heterogeneous Federated Learning with Rolling Sub-Model Extraction

**Ethical Concerns:**

["NO or VERY MINOR ethics concerns only"]

**Final Justification:**

I still think this paper is not good enough to be accepted.

**Limitations:**

The authors could further strengthen limitation section by briefly discussing potential privacy or fairness implications that might arise from their approach, especially concerning the requirement for public datasets, and perhaps outline initial thoughts on how they plan to address the LLM challenges in future work.

**Quality:**

3

**Strengths And Weaknesses:**

**Strength**：

1. This paper addresses a key challenge in real-world FL deployment - model heterogeneity. Effectively supporting model heterogeneity among participating devices enables broader deployment of federated learning across devices with varying resources.
2. The paper is well-written with clear logical organization throughout. Additionally, the authors present the proposed algorithm with clear illustrations and well-structured pseudo-code, making it easy for readers to understand the methodology.
3. The authors perform extensive experiments to demonstrate FedFD effectiveness, including various Non-IID scenarios (α=0.1, 1.0, 10.0), multiple datasets (CIFAR10, CIFAR100, Tiny-ImageNet), and comprehensive ablation studies.



**Weaknesses**：

1. The primary issue with the method's design is how to obtain high-quality public datasets. Several references demonstrate that the correlation between public datasets and local datasets on participating devices significantly impacts model performance[1], and obtaining such datasets is extremely challenging. Additionally, acquiring these datasets to some extent compromises the privacy of users' local data.
2. There's a mismatch between the paper's motivation and experiments. The main purpose of model heterogeneity is to enable devices with different computational resources to participate effectively. Typically, models of varying complexity and architecture (e.g., ResNet, CNN) would be deployed on these devices, but FedFD's evaluation shows very low model heterogeneity, with differences only in channel numbers. This fails to effectively validate the proposed method's effectiveness. Furthermore, some existing methods like HeteroFL and FedRolex have already implemented similar approaches, requiring further discussion on the differences.
3. FedFD lacks theoretical analysis to prove its effectiveness. Additionally, the experimental section uses overly simple models (like ResNet-18) and datasets (like CIFAR). Testing FedFD on a broader range of datasets is necessary.
4. There are inconsistencies in font sizes, for example in lines 270 and 271.



[1]FedZKT: Zero-Shot Knowledge Transfer towards Resource-Constrained Federated Learning with Heterogeneous On-Device Models

---

> ### Author Rebuttal · Authors · 2025-07-29
>
> Thank you for your careful review and valuable comments. In the following, we give point-by-point responses to each comment.
>
> > **W1&Q1.  Concerns about public dataset.**
>
> **R1:** Thank you for this constructive suggestion. Research on federated knowledge distillation can be approached from two main perspectives: (1) the design of distillation algorithms, and (2) the generation of high-quality public datasets with privacy guarantees. Work in category (1), such as FedFusion [1], typically assumes access to high-quality public datasets and focuses on developing distillation algorithms to better leverage such data for performance improvement. In contrast, research in category (2) explores data-free distillation, aiming to synthesize privacy-preserving distillation data. Representative methods such as FedGen[2], FedFTG [3], and DaFKD [4] adopt generative models (e.g., GANs) to generate synthetic data. **These (2)-type methods can often be viewed as add-on components to (1)-type approaches, as they require additional generative models to support data-free distillation.**
>
> **Our proposed method, FedFD, primarily focuses on distillation algorithm design but is also compatible with data-free distillation techniques.** To validate this, we conduct additional experiments where we employ the three aforementioned data-free methods to synthesize distillation data and evaluate the effectiveness of FedFD. Unlike logit-based distillation, even though the quality of synthetic data may not match that of assumed high-quality public datasets, FedFD leverages feature-level outputs to capture richer knowledge, resulting in no significant performance degradation.
>
> | Methods          | Metrics | CIFAR-10 ($\alpha=1.0$) | CIFAR-100 ($\alpha=1.0$) |
> | ---------------- | ------- | ----------------------- | ------------------------ |
> | **FedFD-Public** | Local   | 78.03                   | 52.24                    |
> |                  | Global  | 89.64                   | 60.86                    |
> | FedFD-FedGen     | Local   | 77.12                   | 50.93                    |
> |                  | Global  | 87.99                   | 58.91                    |
> | FedFD-FedFTG     | Local   | 77.04                   | 51.35                    |
> |                  | Global  | 88.11                   | 59.14                    |
> | FedFD-DaFKD      | Local   | 77.58                   | 51.45                    |
> |                  | Global  | 88.97                   | 60.02                    |
>
> Furthermore, to ensure fair comparison in our experiments, the public dataset used for distillation is deliberately chosen to have distributional differences from the training dataset. Specifically, we use {CIFAR-100, Tiny-ImageNet, ImageNet} as the distillation datasets for {CIFAR-10, CIFAR-100, Tiny-ImageNet}, respectively. It is also a common experimental setting in (1)-type works.
>
> > **W2&Q2. Concerns about model heterogeneity setting.**
>
> **R2:**  Thank you for raising this concern. We fully agree with the esteemed reviewer’s observation that the main purpose of model heterogeneity is to enable devices with different computational capabilities to participate effectively. In our experimental setup, we choose HeteroFL as the backbone paradigm for two main reasons:
>
> (1) The channel heterogeneity addressed by HeteroFL is itself a form of model heterogeneity. In this setting, different devices deploy sub-models with similar architectures but varying sizes and computational requirements. Such scenarios are common in practice—for example, **in UAV clusters or joint ventures between subsidiaries with a trusted terminal—where they often follow their own protocol standards that define the set of model architectures to be used, rather than allowing each client to independently choose entirely different model architectures.** Therefore, there is no need to deploy entirely different model architectures that would introduce significant bias and training overhead. In contrast, adapting model based on hardware capabilities is more practical and cost-efficient.
>
> (2) As the first work to explore feature-based federated distillation, we aim to rigorously validate the effectiveness of our method. To do so, it is important to compare against existing state-of-the-art logit-based federated distillation approaches, which typically rely on parameter aggregation to ensure proper convergence. Without such aggregation, these methods tend to struggle since distillation is insufficient to effectively guide the global model aggregation in the early stages of training, it may lead to unstable convergence or even prevent the model from converging properly, resulting in suboptimal performance.. **Therefore, to ensure a fair comparison, we adopt HeteroFL as the backbone paradigm for model aggregation in our experiments.**
>
> **Finally, and most importantly**, in Table 6 of our original paper (We have reused this table from the original content in our rebuttal for your convienience), we have already conducted experiments to address the reviewer's concern regarding models of varying complexity and architecture (e.g., ResNet, CNN). These experiments evaluate the performance of our method alongside the model-agnostic approach FedMD, demonstrating the superiority of feature-based distillation under heterogeneous model settings.
>
> | 50% CNN, 50% ResNet | CIFAR-10     |              | CIFAR-100    |              | Tiny-ImageNet |              |
> | ------------------- | ------------ | ------------ | ------------ | ------------ | ------------- | ------------ |
> | Method              | $\alpha=1.0$ | $\alpha=0.1$ | $\alpha=1.0$ | $\alpha=0.1$ | $\alpha=10.0$ | $\alpha=1.0$ |
> | FedMD               | 67.59        | 53.10        | 30.52        | 24.47        | 21.29         | 19.95        |
> | **FedFD**           | **71.28**    | **59.51**    | **34.93**    | **28.98**    | **30.79**     | **27.96**    |
>
> Regarding the difference between FedFD, HeteroFL, and FedRolex, **federated knowledge distillation can be viewed as an add-on component to parameter aggregation methods, designed to further improve model performance.** Both HeteroFL and FedRolex serve as base parameter aggregation frameworks, while FedFD leverages a public dataset to enhance the performance of the aggregated models. And for logit-based federated distillation methods, they also enhance the global model aggregated in a FedAvg manner.
>
> > **W3&Q3. Concerns about theoratical analysis and simple experimental settings.**
>
> **R3:**  Thank you for raising this concern. Based on our explanation in **R2**:  *federated knowledge distillation can be viewed as an add-on component to parameter aggregation methods, designed to further improve model performance* — we argue that it is not necessary to independently prove the convergence of HeteroFL, which serves as the backbone paradigm in our work. Furthermore, **there is currently no unified or generally applicable theoretical guarantee for the knowledge distillation process**. Prior works also do not provide convergence proofs specifically for the distillation procedure. The theoretical analyses presented in DaFKD [4] and FedGen [2] focus on the convergence of **generative models**, which is beyond the scope of our work, as we do not introduce any new model that requires global aggregation.
>
> Regarding this part of the experimental analysis, we sincerely apologize that our current setup was based on established practices in prior top-tier conference papers, upon which we have already made improvements. Existing works typically use VGG or ResNet family models and conduct evaluations on datasets such as MNIST, Fashion-MNIST, EMNIST, and, in a few cases, CIFAR-10/100. **Building on this, we further extended the evaluation to the Tiny-ImageNet dataset, which is also widely adopted in federated learning research.** We regret that due to the limited rebuttal period, we were unable to collect and analyze more complex, real-world datasets. However, we will expand our experiments in the final version to include larger models and more realistic datasets as suggested.
>
> > **W4&Q4. Concerns about inconsistent font sizes.**
>
> **R4:**  Thank you for this concern. To visually differentiate our method from the backbone paradigm, we **applied the `\small` command** to the “-hetero” label. **We sincerely apologize if this formatting choice caused confusion or did not align with your presentation standards.** We will polish this distinction in a more effective and visually consistent manner in the revised version.
>
> **PS: We have carefully addressed the concerns you raised in our rebuttal. If there are any remaining questions or clarifications needed, please feel free to point them out during the discussion phase. We sincerely hope that our work will receive your recognition following the rebuttal and discussion. We are looking forward to hearing your feedback soon!**
>
> References used in our rebuttal:
>
> [1] Lin T, Kong L, Stich S U, et al. Ensemble distillation for robust model fusion in federated learning[J]. Advances in neural information processing systems, 2020, 33: 2351-2363.
>
> [2] Zhu Z, Hong J, Zhou J. Data-free knowledge distillation for heterogeneous federated learning[C]//International conference on machine learning. PMLR, 2021: 12878-12889.
>
> [3] Zhang L, Shen L, Ding L, et al. Fine-tuning global model via data-free knowledge distillation for non-iid federated learning[C]//Proceedings of the IEEE/CVF conference on computer vision and pattern recognition. 2022: 10174-10183.
>
> [4] Wang H, Li Y, Xu W, et al. Dafkd: Domain-aware federated knowledge distillation[C]//Proceedings of the IEEE/CVF conference on computer vision and pattern recognition. 2023: 20412-20421.

---

> > ### Author Response · Authors · 2025-08-05
> >
> > Dear reviewer zr5J, during this rebuttal period, we have:
> >
> > - Explain the model heterogeneity setting
> > - Add experiments w.r.t combining FedFD with data-free methods
> > - Explain experimental settings and theoretical analysis
> >
> > In this rebuttal, all the main concerns the esteemed reviewers raised are similar, **and our responses have received acknowledgment and recognition from the other two reviewers**. We would love to hear your feedback on our updates and look forward to discussing any remaining concerns you may have. Thank you for your time and consideration.

---

> > > ### Comment · Reviewer_zr5J · 2025-08-07
> > >
> > > Thank you for the detailed responses provided in the rebuttal. However, obtaining such high-quality public datasets is inherently difficult. While data generation via GANs is a potential solution, it may still pose privacy risks by implicitly leaking user information, and GAN training itself is often unstable. Additionally, the work still has limitations in terms of evaluation. Therefore, I will maintain my original score.

---

> > > > ### Author Response · Authors · 2025-08-07
> > > >
> > > > Thanks for your timely reply. As we have already addressed in our rebuttal, data-free federated distillation is a well-recognized and active research topic, specifically designed to tackle the challenge of acquiring suitable distillation data. Regarding the privacy concerns, the related methods we cited [2][3][4] all provide corresponding solutions—for instance, by introducing differential privacy mechanisms, employing more secure generative models, or performing distillation solely on feature-level representations instead of raw data. This is indeed a well-known and classic problem, which other reviewers have also raised. However, they ultimately considered the approach and our explanation **reasonable and acceptable.**
> > > >
> > > > We hope you can reconsider our score based on the reviewers' responses. Please feel free to reach out to us if you have any further questions. **Your recognition and positive adjustment mean a great deal to us.**
> > > >
> > > > Thank you for your valuable suggestions and professional review.
> > > >
> > > > Best wishes,

---

### Note · Authors · 2025-08-13

Dear SACs, ACs, Reviewers,

Thanks for all your effort in reviewing our paper submitted to NeurIPS! Considering the large variance in scores, which may confuse the AC’s judgment, we would like to provide a brief final clarification.

First, this work is the first to investigate the limitations of logit distillation under model heterogeneity and to propose a novel feature distillation approach for federated distillation.

Second, regarding the concern about high-quality datasets, as we addressed in the rebuttal, data-free federated distillation has already been extensively studied. This is not the main focus of our work. Importantly, our method can be directly integrated with existing data-free techniques, which typically use additional generators to produce distillation data. We have also included supplementary validation experiments in the rebuttal to demonstrate this.

Finally, we respectfully request that the AC take into account the issues we previously raised during the rebuttal phase when making the final decision, in order to ensure a fair and accurate review process.

Thank you for your time and effort in managing the review process.

Best regards,

Authors

---

### Decision · Program_Chairs · 2025-09-17

**Decision:**

Accept (poster)

**Comment:**

This paper proposes a feature-based ensemble federated knowledge distillation paradigm to compensate for the knowledge bias arising from heterogeneous models in federated learning. The main technique include introducing orthogonal projection technique to re-parameterize the projection layer to mitigate the knowledge bias issue. Overall, the propose technique is solid and proved to be effective via numerous experiments.

The reviewers raised a number of questions regarding motivation, experimental design and results, lacking of theoretical analysis, and practical deployment of the method. The authors address most concerns by providing more clarifications and experimental results in the rebuttal, which most reviewers are satisfied with. There are a few concerns remained after the rebuttal, including the reasonability of using high-quality public data for training and the lack of adopting data-free methods to avoid the use of public data. While I agree these issues constitute the weakness of the paper, they are also partially resolved in the rebuttal. Two strong-supported reviewers think using public data is widely adopted and should not constitute the main weakness of the paper. Also, I believe the authors have also included supplementary validation experiments in the rebuttal to demonstrate applicability of integrating the method with existing data-free techniques, which somehow addresses the lack of adopting data-free methods issue.

Overall, this paper receives 2 strong supports and 2 borderline reviews. Given the solid method and experimental results, I propose accept of the paper, but request the authors to incorporate the additional clarifications and experimental results in the rebuttal to the final revision.